# The fall—And rise—In hospital-based care for people with HIV in South Africa: 2004–2017

**Evelyn Lauren** [1]*, **Khumbo Shumba**[2], **Matthew P. Fox** [2,3], **William MacLeod** [2,3], **Wendy Stevens**[4], **Koleka Mlisana**[4,5], **Jacob Bor** [2,3☯], **Dorina Onoya**[2☯]

**1** Department of Biostatistics, Boston University School of Public Health, Boston, Massachusetts, United States of America, **2** Department of Internal Medicine, Health Economics and Epidemiology Research Office, School of Clinical Medicine, Faculty of Health Sciences, University of the Witwatersrand, Johannesburg, South Africa, **3** Department of Global Health, Boston University School of Public Health, Boston, Massachusetts, United States of America, **4** School of Laboratory Medicine and Medical Sciences, University of KwaZulu Natal, Durban, South Africa, **5** National Health Laboratory Service, Johannesburg, South Africa

☯ These authors contributed equally to this work.
* elauren@bu.edu

**Data Availability Statement:** Access to primary data is subject to restrictions owing to privacy and ethics policies set by the South African Government. Requests for access to the data can

## Abstract

ART scale-up has reduced HIV mortality in South Africa. However, less is known about trends in hospital-based HIV care, which is costly and may indicate HIV-related morbidity. We assessed trends in hospital-based HIV care using the National Health Laboratory Service (NHLS) National HIV Cohort. Our study included all adults ≥18 years receiving care in South Africa's public sector HIV program from 2004 to 2017 with at least one CD4 count/viral load test in the NHLS database. We compared trends in the number of patients presenting for and receiving HIV care by facility type: hospitals vs. primary care clinics. We then assessed trends and predictors of incident hospitalization, defined as 2 or more hospital-based lab tests taken within 7 days. Finally, we assessed whether trends in incident hospitalizations could be explained by changes in patient demographics, CD4 counts, or facility type at presentation. Data were analyzed on 9,624,951 patients. The percentage of patients presenting and receiving HIV care at hospitals (vs. clinics) declined over time, from approximately 60% in 2004 to 15% in 2017. Risk of hospitalization declined for patients entering care between 2004–2012 and modestly increased for patients entering care after 2012. The risk of hospitalization declined the most in age groups most affected by HIV. Over time, patients presented with higher CD4 counts and were more likely to present at clinics, and these changes explained almost half the decline in hospitalizations. The percentage of HIV care provided in hospitals declined as patients presented in better health and as treatment was increasingly managed at clinics. However, there may still be opportunities to reduce incident hospitalizations in people with HIV.

## Introduction

Since the public-sector roll-out of antiretroviral therapy (ART) in 2004 [1], South Africa's treatment program has grown to be the largest in the world [2], with almost 5.1 million adults

be made via the Office of Academic Affairs and Research at the National Health Laboratory Service through the AARMS research project application portal: http://www.aarms.nhls.ac.za.

**Funding:** DO and JB were supported by grant R01AI152149 from the National Institute of Allergy and Infectious Diseases. NIH had no role in study design, data collection and analysis, decision to publish, or preparation of the manuscript.

receiving ART by 2020 [3]. The expansion of ART coverage has led to substantial reductions in HIV-related morbidity and mortality [4] and has reduced the need for in-patient, hospital-based HIV care. Nevertheless, late presentation for HIV care [5] and gaps in ART adherence [3, 6, 7] contribute to a persistent but avoidable burden of HIV morbidity and hospitalization [8–11].

Hospitalization is often an indicator of severe disease [8, 9]. In South Africa, people living with HIV (PLWH) who are so sick that they need to be hospitalized face six-month mortality rates of 18–31% in the six months after discharge [10, 12, 13]. In-patient HIV care is also expensive, with an average cost upwards of US$1000 (ZAR 18,500) per hospital stay (in 2013 dollars) in South Africa [11, 14, 15]. However, little is known about national trends in hospi-tal-based care for PLWH in South Africa. Previous research has documented trends in HIV care use at specific hospitals [10, 11]. One population-based cohort study showed that the risk of all-cause hospitalization declined dramatically once a person had started ART [16]. In clini-cal cohort studies, patients with high (>95%) ART adherence (compared to lower adherence) were less likely to be hospitalized and had shorter length of stay when they were hospitalized [8, 9]. High adherence is needed to sustain viral suppression and prevent development of drug resistance [17].

Although ART has reduced hospitalizations associated with HIV and AIDS, longer life expectancy of PLWH has increased risk for non-communicable diseases (NCDs) [18–21], including diabetes, hyperglycaemia, and renal insufficiency [21]. These comorbidities among older PLWH are associated with higher healthcare utilization [22], increased hospitalization rates [23], and excess mortality [24].

Tracking trends in hospital-based care for PLWH could help illuminate the lingering bur-den of HIV morbidity in South Africa and inform strategies to reduce HIV morbidity as well as costly inpatient care. As direct data on hospitalizations of PLWH are not available nation-ally, we analyzed data from South Africa's National Health Laboratory Service (NHLS) National HIV Cohort. NHLS conducts all laboratory monitoring for the public sector HIV care and treatment program. We analyzed national trends in laboratory measures of hospital-based care and incident hospitalization from the start of South Africa's ART rollout through 2017.

## Methods

### Study context

When South Africa first rolled out ART, the National Department of Health (NDoH) offered free HIV/AIDS-related care [2] at district and regional hospitals [1]. As demand for ART increased and the prices of medications fell, NDoH started to shift HIV care to PHC settings to ease the burden on hospitals and to improve access to ART in rural areas. In 2010, NDoH rolled out nurse-initiated and managed antiretroviral treatment (NIMART), greatly expanding the number of health workers who could start patients on treatment [25]. The eligibility crite-ria for ART were gradually expanded–from cluster of differentiation 4 (CD4) T-cell count <200 cells/uL to <350 in 2011, to <500 in 2015, and finally to all patients regardless of CD4 count in 2016. Due to the expansion of routine HIV care to PHCs, by 2015, 96% of South Afri-cans resided within 10 km of an ART-providing health facility [26].

### Data sources

Since the start of the HIV treatment program, laboratory monitoring has been used to assess health at clinical presentation (CD4 count), to determine treatment eligibility (until 2016), to measure immune status at ART initiation (CD4 count and VL through 2009, general blood

workup), and to monitor treatment success (CD4 and/or viral load). HIV is managed as an outpatient condition unless patients become very ill. Laboratory testing, including non-HIV-specific blood workups, is also a standard component of inpatient care.

The National Health Laboratory Service (NHLS) provides laboratory and pathology services to over 80% of the national population of South Africa through a national network of laboratories in public health facilities [27]. Since 2004, the NHLS archives all lab test data in a centralized database, NHLS centralized data warehouse (NHLS CDW). Laboratory tests conducted in KwaZulu-Natal province were only integrated into the NHLS CDW starting in 2010.

We conducted a record linkage exercise in collaboration with NHLS to develop and validate a unique patient identifier, transforming the NHLS database into a National HIV Cohort [28]. The linkage approach combined aspects of probabilistic record linkage with network analysis concepts and achieved high accuracy in a validation study with a 1% overmatching rate and 6% undermatching rate relative to manually coded data [29]. This linkage has enabled longitudinal patient-level analyses of all lab-monitored patients in the public sector HIV program [26, 30–32]. We analyzed a de-identified version of the NHLS National HIV Cohort including data from 2004–2017.

### Study population

We included all adults (≥18 years) receiving HIV care in South Africa's public sector HIV program defined as having had at least one CD4 count or HIV viral load (VL) between 2004–2017. All lab tests conducted in public-sector clinics and hospitals were included. (Lab tests at other facilities such as prisons, military bases, and psychiatric facilities were excluded.) For analyses of incident hospitalizations, we further restricted the population to patients with at least one CD4 lab test between 2004 and 2015 to enable 2 years of follow-up for all patients. KwaZulu-Natal joined NHLS in 2010. To maintain consistency over time, KwaZulu-Natal was excluded from all national analyses but included in province-stratified analyses.

### Measures

For each laboratory test available in the NHLS National HIV Cohort we determined the test type, test date, health facility (clinic vs. hospital), test result, and patient demographics (age, sex, province). Possible test types included CD4 count; HIV viral load; alanine aminotransferase, a measure of liver function (ALT); creatinine clearance, a measure of kidney function (CrCl); haemoglobin (Hb); cryptococcal antigen (CrAg); enzyme-linked immunosorbent assay HIV test (ELISA), polymerase chain reaction HIV test (PCR). A facility's status as a hospital or clinic was determined based on classification by the National Institutes of Communicable Diseases (NICD) at NHLS.

For each patient, we identified the date and existence of key events in HIV care. Date of clinical presentation for HIV care and facility where it occurred were defined as the date and location of a patient's first CD4 or viral load test. First documented viral suppression, indicating that the patient was successfully established on HIV treatment, was defined as first viral load <400 copies/mL.

We defined an indicator for whether a patient was "receiving HIV care at a hospital" or "receiving HIV care at a clinic" in a given year by the presence of either a CD4 count or viral load result at that facility type in that year. (Patients could receive care at both clinics and hospitals in the same year.) Finally, we defined "hospitalization" episodes as the presence of multiple lab tests (including CD4/Viral load as well as other blood work-up tests) taken on 2 or more days within a seven-day period at a hospital. Individuals could have multiple hospitalizations. When assessing incident hospitalizations following entry into care or viral suppression,

we excluded hospitalizations occurring within the first 14 days of presentation or viral suppression, as these may have reflected the same care episode. We note that our lab-based measure of incident hospitalization is a proxy for an underlying clinical event (hospital admission) and we cannot rule out the possibility that the measure could capture some outpatient hospital care. To guide interpretation, we assessed whether patients experiencing "incident hospitalization" were in worse health than other patients, as would be expected if our measure captured significant patient morbidity. As described below, incident hospitalization was strongly correlated with worse health across a range of laboratory measures.

## Analyses

**Trends in hospital-based care and incident hospitalizations.** We assessed annual trends in the number (and proportion) of patients presenting for HIV care at a hospital (vs. clinic) and the number (and proportion) of patients actively receiving HIV care at a hospital (vs. clinic), from 2004–2017 nationally.

We also assessed for trends in the number of incident hospitalizations. We estimated Poisson regression models with heteroskedasticity robust standard errors. We hypothesized that changes in hospitalizations due to HIV would track the age-distribution of HIV prevalence and morbidity, which is elevated for people in their 30s and 40s. To assess how the risk of incident hospitalization changed differentially by age, we estimated models interacting age and year. We then computed the relative change in risk of hospitalization for patients presenting for care in 2004 vs. 2015, by age at presentation. We also assessed annual trends in incident hospitalization by age groups: 18–39 and 65+.

**Predictors of hospitalization.** To identify a broader set of risk factors for hospitalization, we assessed the crude and adjusted associations between the risk of incident hospitalization in the two years following clinical presentation and patient characteristics assessed at presentation: age, sex, province, CD4 count at presentation, an indicator of whether the patient presented at a clinic or hospital (facility type), and year of clinical presentation.

**Explaining trends in hospitalizations following clinical presentation.** Using this multi-level modelling framework, we explored several explanations for time trends in incident hospitalizations. To assess the extent to which the trend in hospitalization could be statistically explained by adjusting for each predictor, we estimated the regression models in 4 ways. First, we assessed whether changes in patient demographics (age, sex, province) led to observed secular changes in hospitalizations, given the higher risk of hospitalization as people age and the potential for changing demographics over time. Second, we assessed whether changes in patient health (CD4 count) at time of presentation statistically explained the secular trends. PLWH with low CD4 counts are at much higher risk for opportunistic infection, adverse reactions to ART, and hospitalization. If people sought care at higher CD4 counts over time, then, we hypothesized, the risk of hospitalization would be expected to decline. Third, we assessed the role of facility type at presentation. Patients presenting in hospitals are likely to be in worse health than patients presenting at clinics and may therefore be at higher risk for later hospitalization. (We note that South Africa also shifted routine outpatient HIV care from hospitals to primary health clinics, and patients receiving outpatient care in hospitals may have been more likely to be hospitalized due to the availability of inpatient services on site.) We also modeled these factors jointly, including demographics, CD4 at presentation, and facility type. Comparison of the estimated RRs for "year of presentation" in the crude and adjusted models revealed the extent to which changes in the health and demographics of patients and facility type at presentation statistically explained temporal trends in the risk of hospitalization following clinical presentation. We also assessed whether relaxing ART eligibility criteria played a role by

enabling PLWH to start treatment at higher CD4 counts and reduce person-time spent at lower CD4 counts.

**Trends in hospitalizations among patients established on ART.** HIV treatment regimens have become less toxic over time [33–35]. Since April 1, 2013, South Africa has offered fixed-dose combination (FDC) ART treatment, allowing patients to take just one pill, once per day [36]. These improvements in quality of care, shown to increase retention in care in prior studies [37–40], may have contributed to a reduction in risk of hospitalization [9]. To assess the role of changes in quality of care for patients established on ART, we assessed secular trends in hospitalization from date of first documented viral suppression. We estimated crude and adjusted Poisson models similar to those above. We interpret trends in hospitalization unexplained by patient characteristics as likely reflecting changes in care.

**Changes in the health status of hospitalized patients.** A final factor we considered was the role of supply constraints, which may moderate utilization. As hospitals become less crowded, there may be less pressure to ration hospital care to the sickest patients. As a result, trends in risk of hospitalization could be driven by shifts in available hospital capacity, with risk increasing when hospitals are empty and falling when hospitals are full. We assessed changes in the health status of patients who were hospitalized over time, in order to determine whether changes in the threshold for hospitalization could have led to apparent changes in risk of hospitalization.

## Ethical considerations

Approval for analysis of de-identified data was granted by Boston University's Institutional Review Board (Protocol No. H-31968), Human Research Ethics Committee of the University of the Witwatersrand (Protocol No. M200447) and NHLS Academic Affairs and Research Management System (Protocol No. PR2010539) with a waiver of consent.

A waiver of informed consent was obtained because: the study was not greater than minimal risk; the data were collected previously as part of a laboratory database; the data were de-identified; and the research could not practicably be carried out without the waiver of consent due to the large number of persons in the database and because contacting these persons could introduce new risks including loss of privacy.

## Results

There were 9,624,951 patients 18 years old or older who had any CD4/Viral load lab test between January 2004 and December 2017 (Table 1). Of these, 7,073,255 presented for HIV care at clinics and 2,551,696 presented for care at hospitals. In total 66% of the cohort was female and the median age at entry into care was 34 years old (IQR: 27, 41). At entry to care, 19% had CD4 count of <100 cells/μL, 17% had CD4 count of 100–199 cells/μL, 42% had CD4 count of 200–499 cells/μL, and 22% had a CD4 count of 500+ cells/μL.

The health of patients who were hospitalized was substantially worse than the health of patients who never had a hospitalization event, with respect to their lab result values (Table 2). For hospitalized patients, the median CD4 count was 190 cells/μL (IQR: 73, 350) and the median HIV viral load was 295 copies/mL (IQR: 0, 30288). For patients who were never hospitalized, the median CD4 count was 321 cells/μL (IQR: 197, 480) and the median HIV viral load was 50 copies/mL (IQR: 0, 617). Differences were also found for other lab test types, suggesting that our measure of hospitalization captures meaningful differences in patient morbidity.

Nationally, there was a substantial rise in the number of patients presenting for care at clinics between 2004–2010 and a decreasing trend thereafter (Fig 1a). There was a similar upward trend in patients presenting for care at hospitals during 2004–2005 but the number of patients

**Table 1. Characteristics of the NHLS National HIV Cohort, 2004–2017 (N = 9,624,951).**

| Characteristic | N (%) |
|---|---:|
| Gender | |
| Female | 6,373,951 (66.2%) |
| Age at entry to HIV care | |
| 18–24 | 1,397,942 (14.5%) |
| 25–34 | 3,580,330 (37.2%) |
| 35–44 | 2,464,230 (25.6%) |
| 45–54 | 1,184,317 (12.3%) |
| 55–64 | 437,415 (4.5%) |
| ≥65 | 560,717 (5.8%) |
| CD4 count at entry to HIV care, cells/μL | |
| <50 | 965,020 (10.0%) |
| 50–99 | 820,140 (8.5%) |
| 100–199 | 1,631,427 (16.9%) |
| 200–500 | 4,085,298 (42.4%) |
| >500 | 2,123,066 (22.1%) |
| Facility type at entry to HIV care | |
| Clinic | 7,073,255 (73.5%) |
| Hospital | 2,551,696 (26.5%) |
| Year of entry to HIV care | |
| 2004 | 133,309 (1.4%) |
| 2005 | 332,997 (3.5%) |
| 2006 | 458,315 (4.8%) |
| 2007 | 526,335 (5.5%) |
| 2008 | 642,902 (6.7%) |
| 2009 | 654,692 (6.8%) |
| 2010 | 1,246,828 (13.0%) |
| 2011 | 1,167,360 (12.1%) |
| 2012 | 959,829 (10.0%) |
| 2013 | 772,610 (8.0%) |
| 2014 | 733,411 (7.6%) |
| 2015 | 708,451 (7.4%) |
| 2016 | 680,625 (7.1%) |
| 2017 | 607,287 (6.3%) |
| Province at entry to HIV care | |
| Eastern Cape | 1,045,940 (10.9%) |
| Free State | 542,716 (5.6%) |
| Gauteng | 2,658,989 (27.6%) |
| KwaZulu-Natal | 2,231,953 (23.2%) |
| Limpopo | 776,892 (8.1%) |
| Mpumalanga | 1,037,551 (10.8%) |
| Northern Cape | 144,384 (1.5%) |
| North West | 648,731 (6.7%) |
| Western Cape | 537,795 (5.6%) |
| Hospitalization | |
| ≥1 hospitalizations* | 695,800 (10.6%) |
| Duration of hospitalization** | |
| < = 6 nights | 547,495 (78.7%) |

*(Continued)*

**Table 1.** (Continued)

| Characteristic | N (%) |
|---|---|
| >6 nights | 148,305 (21.3%) |

**Notes:** Hospitalization is defined by proxy as the occurrence of lab tests at a hospital on two different days within the same seven-day period. A hospitalization episode included the sequence of all such tests. Duration of hospitalization was the nights between the first and last test in this episode.

*% denominator for the proportion of PLWH with ≥1 hospitalization is N = 6,536,201, after excluding PLWH accessing care in KZN and entering care after 2015.

** % denominator for duration of hospitalization is out of all patients with any hospitalization.

presenting at hospitals declined consistently between 2005 and 2017. In terms of the total number of patients receiving care, there was a steady increase in the number receiving care at clinics throughout the study period, 2004–2017 (Fig 1b) while hospitals saw a slight bell-shaped curve, reaching a peak in 2010 at 500,000 patients, and remaining relatively stable since 2013 with about 380,000 patients receiving care at hospitals, nationally. The proportion of patients presenting to HIV care and receiving HIV care at a hospital decreased at a roughly similar pace, from approximately 60% in 2004 to approximately 15% in 2017 (Fig 1c). Similar patterns were observed in all provinces (S1 and S2 Figs).

Hospitalizations among PLWH increases substantially between 2004–2010, before slowing down in 2010 (Fig 1D, **black line**). The proportion of hospitalizations occurring at care presentation declines from 72% in 2004 to 27% in 2010 to 16% in 2015, indicating a growing share of PLWH receiving hospital-based care have previously sought HIV care (Fig 1d, **red line**).

Over the period of study, the risk of hospitalization declined the most in adults 25–45 years old, the age groups most affected by HIV. In 2004, the relationship between hospitalization and age was an inverse-U shape, with the share of patients hospitalized within 2 years of presentation at about 15% for people ages 30 to 60 years (Fig 2a). Between 2004 and 2015, risk of hospitalization declined by 40% among adults 25 to 45 years, with lesser declines among young adults and older adults. By 2015, the risk of hospitalization for PLWH in age groups with highest rates of HIV was approximately 8%. We further observed a steady increase in the percentage of hospitalization as patients become older. Risk of hospitalization may have

**Table 2. Laboratory values comparing hospitalized to never hospitalized patients across the 2-year follow up period.**

| Laboratory Test Type | Hospitalized | Never hospitalized |
|---|---|---|
| | *Median (IQR)* | *Median (IQR)* |
| CD4 Lymphocyte Count (CD4) | 190 (73, 350) | 321 (197, 480) |
| HIV Viral Load (VL) | 295 (0, 30288) | 50 (0, 617) |
| Alanine Aminotransferase (ALT) | 28 (17, 53) | 24 (17, 36) |
| Creatinine Clearance (CrCl) | 53 (21, 102) | 92 (66, 114) |
| Haemoglobin (Hb) | 10 (8, 12) | 12 (10, 13) |

**Notes.** *Sample definitions.* Column labeled "hospitalized" reports on all lab results during the two years after clinical presentation for patients who were hospitalized during that time (i.e. had an episode with hospital-based lab tests on at least two different days within a seven day period). Column labeled "never hospitalized" reports on all lab results for all other patients. *Lab test descriptions.* CD4 counts are a measure of immune function. CD4 counts below 200 cells/mm3 are associated with advanced HIV disease. HIV viral loads over 50 copies/mL are considered unsuppressed, VL over 400 copies are elevated, and VL over 1000 copies/mL among patients on treatment indicate potential treatment failure. ALT values above 36 units/L indicate abnormal liver function. CrCl values below 60 mL/min/1.73 m$^2$ denote abnormal kidney function. Hb levels less than 11 g/dL indicate moderate-to-severe anemia.

                                    

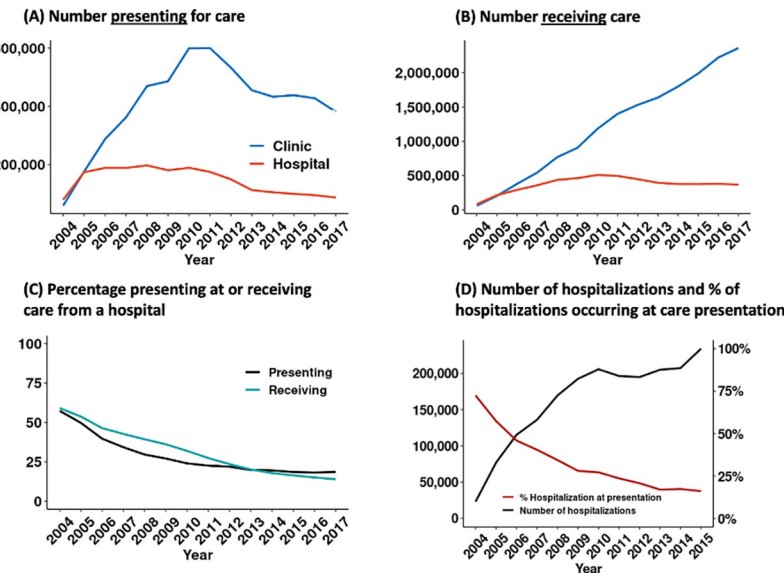

**Fig 1. Trends in hospital-based care for people with HIV in South Africa: 2004–2017. Note**: Figure shows A) number of patients presenting to HIV care by facility; B) number of patients receiving HIV care by facility; C) % of patients presenting to and receiving care at a hospital as a share of total patients receiving HIV care in a clinic or a hospital; D) number of hospitalizations by year of care presentation.

increased over the period among PLWH over 65 years, although the difference was not statistically significant (Fig 2b). In an age-stratified analysis, PLWH over 65 years entering care after 2010 had an elevated risk of hospitalization compared to PLWH 18–39 years (S5 Fig).

We also assessed crude and adjusted trends in the risk that a patient was hospitalized during the two years after presentation (Fig 3a, S1 Table). Risks are expressed relative to the risk of hospitalization for patients presenting in 2004. For all 5 models, the risk ratio of hospitalization after presentation decreases from 2004–2012 and increases from 2013–2015. For the crude model (black line), the risk of hospitalization in 2012 was 0.53 times the risk in 2004, and in 2015 increased modestly to 0.63 times the risk in 2004. Adjusting for patient demographics (green line) did not appear to have a substantial impact on trends in risk of hospitalization. However, CD4 at presentation and facility type did. For the fully adjusted model (yellow line),

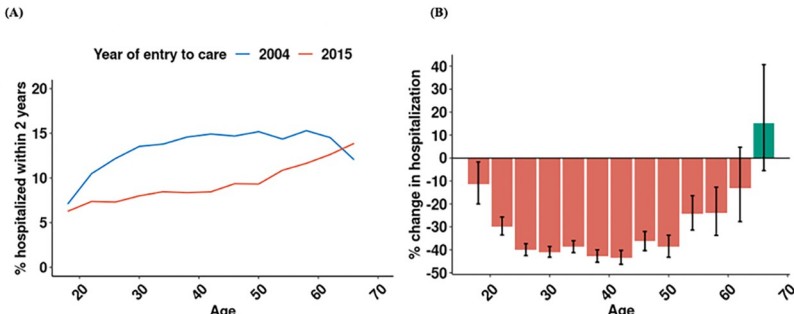

**Fig 2. Hospitalization by age at presentation: 2004 vs. 2015. Note**: Figure shows (A) Percentage of patients hospitalized within 2 years of presentation by age and year presented for care (2004 vs 2015); (B) relative change in the 2-year incidence of hospitalization from 2004 to 2015, stratified by age.

**(A) Risk ratio vs. 2004: hospitalization within 2 years of <u>clinical presentation</u>**

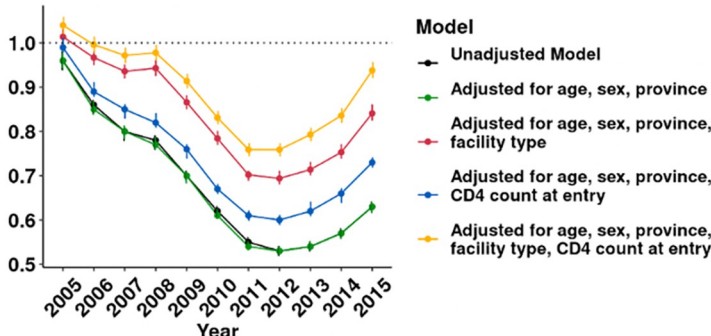

**(B) Risk ratio vs. 2004: hospitalization within 2 years of <u>viral suppression</u>**

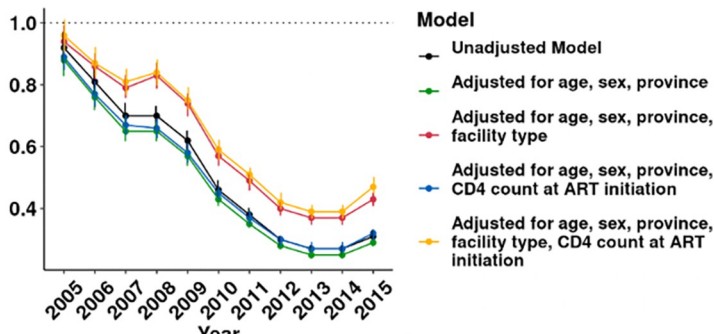

**Fig 3. Crude vs. adjusted annual risk of hospitalization among people accessing HIV care. Note**: Figure shows: (A) risk ratio of hospitalization within 2 years of presentation; (B) risk ratio of hospitalization within 2 years of viral suppression. Risk ratios and 95% CIs are estimated in multilevel Poisson regression models with heteroskedasticity-robust standard errors.

the risk of hospitalization in 2012 was 0.76 times the risk in 2004, and the risk of hospitalization in 2015 is 0.94 times the risk in 2004. In other words, changes in facility type and CD4 count at presentation statistically explained about 50% the decline in hospitalization risk from 2004 to 2012 and over 80% of the difference in risk of hospitalization between 2004 and 2015.

To further investigate the role of patient health at presentation, we assessed the relationship between CD4 count and hospitalization. CD4 count is a well-established predictor of HIV morbidity, and we found that risk of hospitalization was significantly higher at lower CD4 counts at presentation (Fig 4a) and ART initiation (Fig 4b). Over time, patients were more likely to enter care in better health, as indicated by the rightward shift in the distribution of CD4 at presentation (Fig 4c). In 2004, 20% of patients entered care with a CD4 count of 0–49 cells/μL. By 2015, this percentage had dropped to 9%. Expansions of ART eligibility also enabled patients to start ART at higher CD4 counts. Fig 4d shows discrete shifts in the distribution of CD4 at ART initiation coinciding with guideline changes to extend ART eligibility to patients with CD4 200 to 349 cells/uL in mid-2011 and 350 to 499 cells/uL in 2015. Persistent differences in CD4 counts of patients presenting to clinics vs. hospitals are shown in Fig 4e and 4f and facility at presentation may also reflect other unmeasured differences in health (e.g. clinical symptoms).

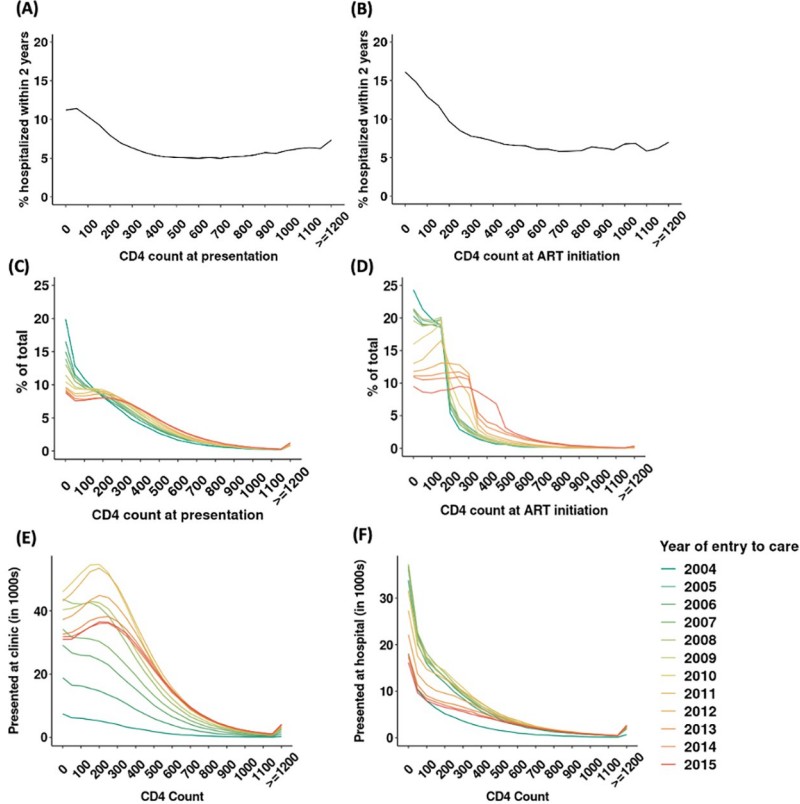

**Fig 4. Hospitalization by CD4 count and changes in the CD4 count distribution over time. Note**: Figure shows (A) percentage of patients hospitalized within 2 years of presentation by value of first CD4 count; (B) percentage of patients hospitalized within 2 years of viral suppression by CD4 count at treatment initiation; (C) distribution (histogram) of CD4 count values at presentation, stratified by year; (D) distribution of CD4 count values at treatment initiation, stratified by year. The last CD4 count before initiation was taken as the initiating CD4 count. Figure also shows the number of patients presenting at (E) clinics and (F) hospitals, by CD4 count and year.

Changes in hospitalization risk were not only due to improvements in health at the time of care-seeking and ART initiation. Patients established on treatment also saw large reductions in hospitalization risk during the study period. The risk of hospitalization after viral suppression decreased by 70% from 2004–2014, before a slight uptick in 2015 (Fig 3b, S2 Table). Adjusting for patient characteristics at presentation explained only 20% of this decline. Other factors– such as the rollout of less toxic medications or increased support for adherence and retention– may have played a role in falling risk of hospitalization for PLWH on ART.

While risk of hospitalization declined for most of the study period, it increased for patients entering care after 2012. After adjusting for changes in patient characteristics (Fig 3a, yellow line), the risk of hospitalization was nearly as high in 2015 as it was in 2004. Whereas residual (unmodeled) factors contributed to a decline in hospitalization risk from 2004 through 2012, residual factors also contributed to a nearly equal rise in hospitalization risk thereafter.

To understand the impact of these trends on total numbers of hospitalizations, we assessed the number of hospitalizations that would have been expected for patients presenting each year 2011–2015, based on changes in patient characteristics (age, sex, province, health facility at presentation, and CD4 count at entry to care) and the estimated associations of those characteristics with hospitalization risk (Fig 5a). Based on changes in patient characteristics,

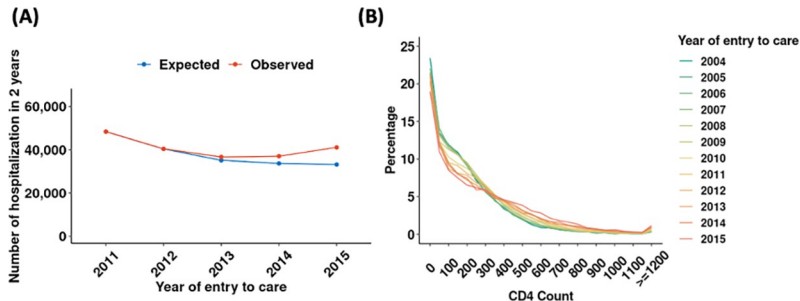

**Fig 5.** (A) Hospitalization within 2 years of presentation, 2011–2015: expected vs. observed; (B) CD4 count at time of hospitalization. **Note:** Figure shows (A) observed number of hospitalizations within 2 years of presentation in red and expected number of hospitalizations within 2 years of presentation based on changes in patient demographics, CD4 counts at presentation, and facility type at presentation in blue; (B) distribution of CD4 counts among patients hospitalized, with patients more likely to be hospitalized at higher CD4 counts in recent years. See S3 Fig for trends in other lab results at hospitalization.

hospitalizations were expected to decline from 2011–2015 (blue line). However, the observed data (red line) show an increase in total number of hospitalizations for patients entering care in 2014 and 2015. We conducted further investigations to check if the shift in trend was an artifact of how hospitalization was defined in this paper. However, we did not observe any notable shift in how tests were administered, in the demographics of patients hospitalized, and in the facility classifications of hospitals and clinics for patients entering care in 2004–2015 (S3 Table). Hence, the increase in hospitalization is unlikely to be due to issues with the dataset and definitions.

While potentially concerning, these findings do not necessarily indicate an increase in HIV morbidity. There were marked improvements in the values of CD4 counts (Fig 5b) and other laboratory measures (S3 Fig) among patients hospitalized in the years 2015–2017, suggesting a lower bar for hospitalization in this later period. Additionally, nearly the entire rise in hospitalization occurred among patients who initially presented for HIV care at hospitals (S4 Fig), suggesting that changes in clinical procedures within hospitals, not rising morbidity, may explain the apparent rise in hospitalizations.

## Discussion

We analyzed trends in hospital-based care and risk of hospitalization among PLWH in South Africa using laboratory data from the NHLS National HIV Cohort, 2004–2017. We found that the share of patients presenting at hospitals and receiving hospital-based care declined over time. Risk of hospitalization also decreased from 2004–2012, before increasing modestly from 2012–2015. Our results are consistent with smaller-scale analyses linking ART to reductions in hospital-based care [41–44], and they illustrate how national scale-up of ART has affected hospital-based HIV care in South Africa [14, 45].

The decline hospitalization risk occurred primarily among persons 30–49 years old, who saw reductions of approximately 40% from 2004–2015. This age group constitutes the majority of HIV patients and may drive the declining trend in the proportion of patients accessing care at hospitals over time. On the other hand, risk of hospitalization among older patients rose from 2004–2015. Older PLWH may be hospitalized due to other reasons, e.g., non-communicable diseases [46–49], and further research will be needed to identify whether rising hospitalization of the elderly reflects improved access to care or an increase in morbidity.

Several factors contributed to the secular decline in hospital-based HIV care and hospitalization in South Africa. There was a consistent increase in ART coverage throughout the period of study. As prior studies have established, ART reduces morbidity and risk of hospitalization for PLWH [44, 50]. As the pool of PLWH with advanced disease started treatment (or died), those PLWH who remained were on average healthier and often entered care at higher CD4 counts before they had experienced serious morbidity [51–54]. Expansions of ART eligibility also enabled patients who presented at higher CD4 counts to start ART without delay. When the ART program was first implemented, treatment was available only at hospitals and major clinics. The era 2007–2010 saw a massive expansion in the number of HIV treatment sites [2]. As such, the decrease observed in the number of people initiating care in hospitals from 2010 to 2013 likely reflects, in part, decentralization of HIV care. Our regression models indicate that improvements in patient CD4 counts at presentation and the increase in care-seeking at clinics statistically explained almost half of the decline in hospitalization risk.

Our analysis reveals several successes of the national HIV program, but also some causes for concern. Even as fewer new patients entered care towards the end of the study period, a considerable proportion still presented with very low CD4 counts, consistent with findings from previous studies [54]. Low CD4 counts at entry to care are associated with a higher probability of hospitalization in these and other data [55–57]. Additionally, over time a growing share of hospital-based care was provided for PLHIV who previously entered care and may have disengaged from care or experienced adherence challenges. Previous research found that HIV-related hospitalization remains common despite the success of ART scale-up in South Africa [10, 11]. We observed that the number of patients receiving care in hospitals flat-lined after 2013. At the same time, there was no increase in the number of hospital beds in South Africa [58], according to World Bank statistics, raising concern that hospitals may remain overburdened and under pressure.

Lastly, we observed an increasing risk of incident hospitalization from 2013 onwards. This rise was particularly pronounced when adjusting for patient CD4 count and facility type at presentation. We believe it is unlikely that this rising trend reflects a substantial increase in patient morbidity–which has not been reported elsewhere. We conducted several robustness checks and were able to rule out explanations related to the dataset or definitions. One possible, if speculative, explanation is that the success of the country's treatment program, with improvements in patient health and declining numbers entering HIV care, may have reduced congestion pressures at hospitals. Patients hospitalized in 2015–2017 were hospitalized while in better health than previously, at least according to available laboratory measures. It is possible that hospitals had to ration HIV care to the sickest patients early in the study period, but later were able to allocate resources to marginally healthier patients leading to higher hospitalization rates as patients were able to access the care they needed. Further investigation and alternate data on hospital congestion are needed to test this hypothesis and investigate other explanations for the rise in hospitalization after 2013.

There are several strengths to this study including the large sample size, the national-level analyses, and the longitudinal nature of our data. Still, our study had several limitations. First, the NHLS database only included laboratory results and did not contain other information often available in clinical patient records, such as referral notes and hospital admission or discharge indicators. Hence, although we were able to accurately distinguish between laboratory tests taken in hospital vs. clinics, we were limited to indirect methods to infer hospitalization events. Our laboratory-based measure of hospitalization likely includes some misclassification. However, we did find that it was strongly correlated with patient morbidity. We were also limited in the types of laboratory tests available for inclusion in the study. Second, as with any large linked administrative database, our data are not impervious to linkage errors. The NHLS

National HIV Cohort obtained high sensitivity and PPV in a validation study [29] and thus our findings should be fairly robust to linkage error. Third, although reductions in hospitalization are most likely due to reductions in patient morbidity with ART scale-up, we cannot rule out the possibility that a decline in hospitalizations resulted from referral failures within the health system [59–62]. Fourth, KwaZulu-Natal was excluded from all national-level analyses as data were only available starting in 2010. In provincial breakdowns of individuals presenting and receiving HIV care (S1 and S2 Figs), KZN follows similar trends as other provinces post-2010. Fifth, due to limitations in data availability, we were unable to analyze policy changes beyond 2017, such as the dolutegravir (DTG) rollout, which may affect hospitalizations. However, the period of study captures most major HIV policy changes in South Africa, including the ART rollout, the introduction of NIMART, the expansion of CD4 count thresholds for ART eligibility, introductions of new ART regimens, and improved diagnostics for TB. Since 2017, there have been few major changes to HIV care, the rollout of DTG notwithstanding. Nevertheless, we caution against extrapolating these results to today's ART program.

## Conclusion

We analyzed trends in hospital-based care among HIV patients in South Africa. We observed a decline in the share of HIV care provided in hospitals over time, as well as a decline in hospitalizations over time. The success of HIV care decentralization through NIMART, better health at presentation and treatment initiation, and improvements in quality of ART care all contributed to the decline. However, the total number of PLWH receiving hospital-based care has flat-lined since 2013. A considerable proportion of PLWH still present with very low CD4 counts; hospitalization of older PLWH remains high; and risk of incident hospitalization after clinical presentation was increasing at the end of the study period. Even in South Africa's mature HIV treatment program, there may still be opportunities to engage PLWH in care earlier in HIV infection, to reduce HIV-related morbidity, and to ensure PLWH get the hospital-based care they need when they need it.

## Supporting information

**S1 Checklist. STROBE statement.**
(DOCX)

**S1 Table. Risk ratio of hospitalization in 2 years after presentation by year of entry to care.**
(DOCX)

**S2 Table. Risk ratio of hospitalization in 2 years after viral suppression by year of entry to care.**
(DOCX)

**S3 Table. Potential factors driving hospitalization rates.**
(DOCX)

**S1 Fig. Number of patients presenting to HIV care by facility and province.**
(DOCX)

**S2 Fig. Number of patients receiving HIV care by facility and province.**
(DOCX)

**S3 Fig. Trends in lab test results at first hospitalization.**
(DOCX)

**S4 Fig. Percentage of patients hospitalized within 2 years after presentation and viral suppression by facility at entry to care.**
(DOCX)

**S5 Fig. Percentage of patients hospitalized within 2 years after presentation by age group (18–39, 40–64, 65+).**
(DOCX)

## Author Contributions

**Conceptualization:** Evelyn Lauren, Khumbo Shumba, Matthew P. Fox, William MacLeod, Jacob Bor, Dorina Onoya.

**Data curation:** Khumbo Shumba, Matthew P. Fox, William MacLeod, Wendy Stevens, Koleka Mlisana, Jacob Bor, Dorina Onoya.

**Formal analysis:** Evelyn Lauren, Khumbo Shumba, Matthew P. Fox, William MacLeod, Jacob Bor, Dorina Onoya.

**Funding acquisition:** Jacob Bor.

**Investigation:** Jacob Bor, Dorina Onoya.

**Methodology:** Evelyn Lauren, Khumbo Shumba, Matthew P. Fox, William MacLeod, Jacob Bor, Dorina Onoya.

**Project administration:** Koleka Mlisana, Jacob Bor, Dorina Onoya.

**Resources:** Wendy Stevens, Jacob Bor, Dorina Onoya.

**Software:** Jacob Bor, Dorina Onoya.

**Supervision:** Jacob Bor, Dorina Onoya.

**Validation:** Khumbo Shumba, Wendy Stevens, Koleka Mlisana, Jacob Bor, Dorina Onoya.

**Visualization:** Jacob Bor, Dorina Onoya.

**Writing – original draft:** Evelyn Lauren, Khumbo Shumba, William MacLeod, Jacob Bor, Dorina Onoya.

**Writing – review & editing:** Evelyn Lauren, Khumbo Shumba, Matthew P. Fox, William MacLeod, Wendy Stevens, Koleka Mlisana, Jacob Bor, Dorina Onoya.

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
