## [Editor Report · Decision Letter 0]

15 Jun 2023

PGPH-D-23-00923

The fall -- and rise -- in hospital-based care for people with HIV in South Africa: 2004-2017

Dear Dr. Lauren,

Thank you for submitting your manuscript to PLOS Global Public Health. After careful consideration, we feel that it has merit but does not fully meet PLOS Global Public Health’s publication criteria as it currently stands. Therefore, we invite you to submit a revised version of the manuscript that addresses the points raised during the review process.

EDITOR: Please insert comments here and delete this placeholder text when finished. Be sure to:

This is an excellent paper, very clearly presented and an easy read.Prior to publication, it would benefit from further consideration of the factors that contributed to the later rise in hospitalisation.  Were factors such as proximity to hospitals, urban versus rural, change in hospital guidelines for admission considered? If so these need to be reflected. I would also like to see some data on the patient characteristics of the hospitalised versus non-hospitalised patients beyond age. What other characteristcs were considered and found to be relevant or not. A couple of paragraphs address this in the discussion section will be good 

We look forward to receiving your revised manuscript.

Kind regards,

Ebere Okereke, MBBS, DTM&H, MSc (PH), FFPH

Academic Editor

Journal Requirements:

b. If any authors received a salary from any of your funders, please state which authors and which funders.

3. Please provide separate figure files in .tif or .eps format only and remove any figures embedded in your manuscript file. Please also ensure all files are under our size limit of 10MB.

4. We notice that your supplementary figures and tables are included in the manuscript file. Please remove them and upload them with the file type 'Supporting Information'. Please ensure that each Supporting Information file has a legend listed in the manuscript after the references list.
---

## [Decision Letter · Decision Letter 1]

14 Mar 2024

PGPH-D-23-00923R1

The fall -- and rise -- in hospital-based care for people with HIV in South Africa: 2004-2017

Dear Dr. Lauren,

Thank you for submitting your manuscript to PLOS Global Public Health. After careful consideration, we feel that it has merit but does not fully meet PLOS Global Public Health’s publication criteria as it currently stands. Therefore, we invite you to submit a revised version of the manuscript that addresses the points raised during the review process.

We look forward to receiving your revised manuscript.

Kind regards,

Syed Shahid Abbas, MBBS, MPH, Ph.D.

Academic Editor

Journal Requirements:

Additional Editor Comments (if provided):

Reviewers' comments:

Reviewer's Responses to Questions

**Comments to the Author**

1. If the authors have adequately addressed your comments raised in a previous round of review and you feel that this manuscript is now acceptable for publication, you may indicate that here to bypass the “Comments to the Author” section, enter your conflict of interest statement in the “Confidential to Editor” section, and submit your "Accept" recommendation.

Reviewer #1: (No Response)

Reviewer #2: (No Response)

Reviewer #3: (No Response)

2. Does this manuscript meet PLOS Global Public Health’s publication criteria? Is the manuscript technically sound, and do the data support the conclusions? The manuscript must describe methodologically and ethically rigorous research with conclusions that are appropriately drawn based on the data presented.

Reviewer #1: (No Response)

Reviewer #2: Yes

Reviewer #3: Yes

3. Has the statistical analysis been performed appropriately and rigorously?

Reviewer #1: Yes

Reviewer #2: Yes

Reviewer #3: Yes

4. Have the authors made all data underlying the findings in their manuscript fully available (please refer to the Data Availability Statement at the start of the manuscript PDF file)?

Reviewer #1: No

Reviewer #2: No

Reviewer #3: No

5. Is the manuscript presented in an intelligible fashion and written in standard English?

Reviewer #1: (No Response)

Reviewer #2: Yes

Reviewer #3: Yes

6. Review Comments to the Author

Reviewer #1: I have several minor queries and comments listed below. There are no page numbers in the review package, so I have used the Heading, sub-heading and paragraph as a guide to where the comments were raised.

Comments:

Introduction, paragraph 2:

As this is a South African national cohort, can the authorships add a ZAR equivalent value for the cost of inpatient HIV care?

Methods: Study population:

Can the authors expand on the sentence “To maintain consistency, KwaZulu-Natal was excluded from all national-level analyses, because the province joined NHLS in 2010.”?

What analyses were KwaZulu-Natal excluded from?

Does the NHLS National HIV Cohort (Table 1) include individuals from KwaZulu-Natal?

It may be helpful to include a breakdown of the number of individuals from each province of South Africa.

KwaZulu-Natal is the third-largest province in South Africa by population count. As per the latest estimates, approximately 2.1 million people living with HIV are located in KwaZulu-Natal out of the country’s total ~7.9 million PLWH. Excluding KwaZulu-Natal is a significant limitation of the study. This needs to be noted.

Methods: Analyses: Trends in hospital-based care and incident hospitalizations:

Do the authors know when the change in the ART regimens took place in South Africa?

Please add a reference & citation on the lessened toxicity of the one-pill regimen.

Methods: Analyses: Changes in the health status of hospitalized patients:

How was the threshold for hospitalization determined? As this is a laboratory test-record-based dataset, were indicators of hospitalizations recorded in the lab records?

Results: Paragraph 2:

On the statement “patients who were never hospitalized”, please add additional clarification, the patients were never hospitalized for HIV related treatments. The patients could have been hospitalized for other conditions, where a baseline HIV test was performed.

Results: Paragraph 12:

Two points here, the first one is a bit more philosophical. “Additionally, nearly the entire rise in hospitalization occurred among patients who presented for care at hospitals.” – one can not be hospitalized if they are not at a hospital. Can the authors add some clarity here, can a patient be referred for hospitalization by a Clinic, without the hospital having to repeat the HIV tests? Second comment, do the authors know how many patients first presented to a Clinic for HIV care, and subsequentially were referred to a Hospital for continued HIV care?

Discussion: Paragraph 4:

“Another potential explanation is that the success of the country’s treatment program and declining numbers of new patients entering care may have reduced congestion pressures at hospitals.” – I have two issues with this statement. Firstly, it directly contradicts the authors’ prior statement a few sentences earlier “Thus, there is a concern that hospitals remain overburdened and under pressure.”. Secondly, although the improvement of HIV care over time in South Africa indeed has had significant positive impacts on the health and wellbeing of PLWH. However, proposing that the increase in HIV-related hospitalization is due to increased hospital capacity is unfounded as no evidence pointing to the increased hospital capacity in South Africa is presented in the manuscript. This is also a siloed view of public care in South Africa. HIV care is but one of many disciplines of healthcare. The demand for the public healthcare system in South Africa continues to increase year-on-year. Therefore, this statement suggests a false viewpoint, unless the authors can present evidence for the contrary.

Grammar, abbreviations and consistency:

1. “In-patient” or “inpatient” & “out-patient” or “outpatient”, please pick one and be consistent.

2. Methods: Study context: “PHC”, please expand. This is the first use of this abbreviation.

3. Methods: Study context: Please add units for CD4.

4. Methods: Data source: “VL”, please expand. First use of the abbreviation and please be consistent with its use.

5. Results: paragraph 1: “In total 66%..” please add a comma: “In total, 66% …”.

Reviewer #2: Thank you for the opportunity to review this paper. Although its labelled revision 1, I was seeing it for the first time. The paper was generally well written and addresses an important issue around continuing HIV related morbidity and hospitalization in the era of widely available HAART. The so what of the paper could be better handled/ discussed. The paper didn't have page numbers or line numbers which made reviewing and referencing sections difficult. I have made more detailed comments below:

Abstract

- the authors refer to their database as novel. I don't agree. This NHLS cohort created from record linkage has been around for a while - as early as 2015/2016 and has been used to publish on different HIV related outcomes . Maybe its the first time they have used it to look at hospitalization as an outcome, but its certainly not novel

- why 2017? the data is already dated and so many events that could potentially affect hospitalizations among HIV infected have happened i.e. COVID-19 pandemic and DTG roll-out

- the outcome was determined from the variable facility type i.e. clinic vs hospital. How did the authors handle hospital based out-patient , ambulatory HIV services which may not be labelled clinics but function as such

Introduction

- the authors didn't mention about NCD related comorbidities that are becoming increasingly common among individuals aging on ART as a potential factor in increasing hospitalization of PLHIV over time

- the authors refer to this database as novel- please see abstract for comment on this

Methods

Under data sources - the authors refer to the CD4 count as a measure of health at ART start . I think its best to say as a measure of immune status. Health is too non-specific

Under measures - why an ELISA for HIV when the bulk of HIV diagnoses are done by rapid HIV testing. What was the rationale for the ELISA. The same question applies to the PCR?

Why weren't TB related measures included when disseminated TB is the commonest opportunistic infection among PLHIV. Measures of disseminated TB could have been explored - eg CSF samples for TB,

As markers of incident hospitalization- why only blood work-up?

Also was there an attempt to look at drug resistance data since ART failures are an important factor contributing to hospitalisations

Analyses

- comparing risk of hospitalization in 2004 vs 2017 doesn't address the query I had about increasing risk of hospitalization with increasing age in the same individuals over time. Is it possible to include this type of analysis

Results

- the authors found quite large numbers of people who had a CD4 count/ viral load and therefore assumed to have initiated HIV care or started ART. Was there an attempt to check with other data sources whether these numbers are plausible. SA has an estimated 5.8 million people on ART out of 7.4 million PLHIV. Does this mean over the years 2 million people plus have died from HIV?

- in paragraph 5 of the results section, the authors present results on children when in the method state that they only included adults in the analysis.

Discussion

the so what of the paper is could be strengthened. In the discussion, the contribution of disengagement from care and drug resistance to hospitalization has not been discussed. As early as 2012/2013 a study from Cape Town found that half of HIV related admissions were among people who had disengaged form care

Any recommendations for policy or practice?

Reviewer #3: General

• This is a clear and well-written manuscript assessing temporal changes in hospital- vs primary care clinic-based care and incident hospitalisation within a very large and highly representative in South Africa from 2004 to 2017. While the data is no longer very recent, the authors position the importance of this topic well in the introduction, highlighting implications of hospitalisation both for individual health and the health system. For the most part, I have only minor comments for consideration by the authors.

Methods:

• Subsection “data sources”, paragraph 3: Please consider whether the word “anonymized” is accurate in this case or whether pseudonymized, deidentified etc. might be more appropriate (under “ethical considerations”, the term “de-identified” is used). Though often used differently, “anonymized” implies that no linkage is even theoretically possible – i.e. that all direct and indirect links between identifiers and identifying information have been removed.

• Subsection “measures”, paragraph 1: The abbreviation “CD4” is used repeatedly before being introduced here.

• Subsection “measures”, paragraph 3: “We excluded hospitalizations occurring within the first 14 days of presentation or viral suppression, as these may have reflected the same care episode.” This is not entirely clear to me and appears to by default exclude hospitalization at presentation. If I understand correctly, the authors may consider clarifying that this manuscript addressed incident hospitalisation among people already in care for HIV (/in pre-ART care), not incident hospitalisation related to late presentation. This appears to be contradicted by the tested association between health (CD4 count) at presentation and risk of hospitalization, so I apologize to the authors if I misunderstood this.

• The authors could briefly mention how missing data and multiple hospitalizations were handled.

Results:

• Table 1: In the caption, “20018” should probably read “2017”. Is it correct that there was no missing data for the displayed variables (the authors may wish to briefly address missing data in the methods)? For hospitalizations up to or more than 6 nights, please indicate more clearly that the denominator is all people with hospitalization, not all people in the dataset. I suggest additionally showing the proportion of all people in the dataset who had a hospitalization.

• Paragraph 5 (description of figure 2): While it is clear from context and is likely known to most readers, I suggest stating explicitly which age groups are considered “most affected by HIV”.

• Figure 4 C-F, Figure 5B: I find these panels hard to read especially as similar colours are used for non-consecutive years. Perhaps a colour gradient could be used such that consecutive years have more similar colours.

• “After adjusting for changes in patient characteristics (Figure 2A, yellow line), the risk of hospitalization was nearly as high in 2015 and in 2004” I believe this should be Figure 3A.

Discussion:

• “Our regression models indicate that CD4 count at care presentation and facility at presentation, explained almost half of the trend in hospitalization risk.” Here and elsewhere, “explained” appears to me to imply causality. However, it is not clear to me that the methods justify causal statements rather than statements of association. Regarding facility at presentation, do the authors believe that presentation at a clinic is directly causative of a lower risk of hospitalization (e.g. because “patients receiving outpatient care in hospitals may have been more likely to be hospitalized due to the availability of inpatient services on site” as noted in the methods), or that both presentation at a clinic and lower risk of hospitalization have a shared underlying cause (e.g. better health beyond what can be adjusted for with CD4 count)? In the former case, there could be concern that clinic presentation reduces the likelihood of receiving a required hospitalization.

• The authors draw some conclusions for HIV care today from this data – e.g., that “there may still be opportunities for earlier case-finding and reduction in incident hospitalizations in PLWH”. While I certainly agree, it may be appropriate to briefly note as a limitation that follow-up ended in 2017 and extrapolations to today’s ART programme are difficult.

7. PLOS authors have the option to publish the peer review history of their article (what does this mean?). If published, this will include your full peer review and any attached files.

**Do you want your identity to be public for this peer review?** For information about this choice, including consent withdrawal, please see our Privacy Policy.

Reviewer #1: No

Reviewer #2: No

Reviewer #3: No

---

## [Decision Letter · Decision Letter 2]

26 Jul 2024

PGPH-D-23-00923R2

The fall -- and rise -- in hospital-based care for people with HIV in South Africa: 2004-2017

Dear Dr. Lauren,

Thank you for submitting your manuscript to PLOS Global Public Health. After careful consideration, we feel that it has merit but does not fully meet PLOS Global Public Health’s publication criteria as it currently stands. Therefore, we invite you to submit a revised version of the manuscript that addresses the points raised during the review process.

Please address the minor request from Reviewer #2.

We look forward to receiving your revised manuscript.

Kind regards,

Marianne Clemence

Staff Editor

Journal Requirements:

Additional Editor Comments (if provided):

Reviewers' comments:

Reviewer's Responses to Questions

**Comments to the Author**

1. If the authors have adequately addressed your comments raised in a previous round of review and you feel that this manuscript is now acceptable for publication, you may indicate that here to bypass the “Comments to the Author” section, enter your conflict of interest statement in the “Confidential to Editor” section, and submit your "Accept" recommendation.

Reviewer #1: All comments have been addressed

Reviewer #2: All comments have been addressed

2. Does this manuscript meet PLOS Global Public Health’s publication criteria? Is the manuscript technically sound, and do the data support the conclusions? The manuscript must describe methodologically and ethically rigorous research with conclusions that are appropriately drawn based on the data presented.

Reviewer #1: Yes

Reviewer #2: Yes

3. Has the statistical analysis been performed appropriately and rigorously?

Reviewer #1: Yes

Reviewer #2: Yes

4. Have the authors made all data underlying the findings in their manuscript fully available (please refer to the Data Availability Statement at the start of the manuscript PDF file)?

Reviewer #1: Yes

Reviewer #2: No

5. Is the manuscript presented in an intelligible fashion and written in standard English?

Reviewer #1: Yes

Reviewer #2: Yes

6. Review Comments to the Author

Reviewer #1: Dear Authors, Thank you for your robust responses to my comments. I am satisfied that all my comments have been answered. I have no further comments.

Reviewer #2: Thank you for the opportunity to review the revised manuscript. The authors did a good job addressing the comments I had. A few minor things to consider

i) NICD in line 136 in the revised manuscript should be National Institute for Communicable Diseases and not clinical diseases

ii) in the comparison 18- 39 vs 65+, why was 40- 64 left out of the discussion and supplementary chart?

7. PLOS authors have the option to publish the peer review history of their article (what does this mean?). If published, this will include your full peer review and any attached files.

**Do you want your identity to be public for this peer review?** For information about this choice, including consent withdrawal, please see our Privacy Policy.

Reviewer #1: No

Reviewer #2: No

---

## [Decision Letter · Decision Letter 3]

13 Aug 2024

The fall -- and rise -- in hospital-based care for people with HIV in South Africa: 2004-2017

PGPH-D-23-00923R3

Dear Ms. Lauren,

We are pleased to inform you that your manuscript 'The fall -- and rise -- in hospital-based care for people with HIV in South Africa: 2004-2017' has been provisionally accepted for publication in PLOS Global Public Health.

Best regards,

Claudia P. Cortes, MD

Academic Editor

the authors have incorporated all the suggestions of the reviewers in the previous rounds and in my opinion the current quality of the manuscript is in a position to be accepted.

I have only one doubt:

in line #60 it says "average cost upwards of US$1000 (ZAR 18,500) per hospital stay (in 2013 dollars)".

please confirm that it is the value of the dollar of the year 2013?! or is it a typo and refers to the year 2023? (should be at least 2023 or even 2024 exchange rate)

this is the only point to clarify before accepting the publication.

Reviewer Comments (if any, and for reference):

Reviewer's Responses to Questions

**Comments to the Author**

1. If the authors have adequately addressed your comments raised in a previous round of review and you feel that this manuscript is now acceptable for publication, you may indicate that here to bypass the “Comments to the Author” section, enter your conflict of interest statement in the “Confidential to Editor” section, and submit your "Accept" recommendation.

Reviewer #2: All comments have been addressed

2. Does this manuscript meet PLOS Global Public Health’s publication criteria? Is the manuscript technically sound, and do the data support the conclusions? The manuscript must describe methodologically and ethically rigorous research with conclusions that are appropriately drawn based on the data presented.

Reviewer #2: Yes

3. Has the statistical analysis been performed appropriately and rigorously?

Reviewer #2: Yes

4. Have the authors made all data underlying the findings in their manuscript fully available (please refer to the Data Availability Statement at the start of the manuscript PDF file)?

Reviewer #2: Yes

5. Is the manuscript presented in an intelligible fashion and written in standard English?

Reviewer #2: Yes

6. Review Comments to the Author

Reviewer #2: Thank you for the opportunity to review this revised manuscript. The authors have addressed all the comments I had and I have no further queries

7. PLOS authors have the option to publish the peer review history of their article (what does this mean?). If published, this will include your full peer review and any attached files.

**Do you want your identity to be public for this peer review?** For information about this choice, including consent withdrawal, please see our Privacy Policy.

Reviewer #2: No
